# The Role of Vitamins in Oral Potentially Malignant Disorders and Oral Cancer: A Systematic Review

**DOI:** 10.3390/jpm13101520

**Published:** 2023-10-23

**Authors:** Jewel Kai Lin See, Xinyao Liu, Federica Canfora, Caroline Moore, Michael McCullough, Tami Yap, Rita Paolini, Antonio Celentano

**Affiliations:** 1Melbourne Dental School, The University of Melbourne, 720 Swanston Street, Carlton, VIC 3053, Australia; jewelkailins@student.unimelb.edu.au (J.K.L.S.); xinyl@student.unimelb.edu.au (X.L.); federica.canfora@unina.it (F.C.); moore@unimelb.edu.au (C.M.); m.mccullough@unimelb.edu.au (M.M.); tspyap@unimelb.edu.au (T.Y.); rita.paolini@unimelb.edu.au (R.P.); 2Department of Neuroscience, Reproductive Sciences and Dentistry, University of Naples Federico II, 5 Via Pansini, 80138 Naples, Italy

**Keywords:** oral potentially malignant disorders, oral cancer, vitamins, trace element

## Abstract

Background: Micronutrients are vital for general and oral health, and their potential anti-cancer properties are documented. We explore beneficial vitamins for oral potentially malignant disorders (OPMDs) and oral cancer (OC), assessing the therapeutic impacts of essential vitamin supplementation. Methods: We systematically review evidence on vitamin supplementation’s therapeutic effects for OPMDs and OC. Relevant studies were identified through comprehensive searches of MEDLINE, Evidence-Based Medicine, and Web of Science until 16 May 2023. All studies underwent risk of bias using criteria modified from the Office of Health Assessment and Translation (OHAT) tool. Results: We analysed 80 papers. Vitamin K, studied in vitro, shows promising therapeutic potential. Vitamin C, investigated in vivo (animals and humans), demonstrated mixed animal results and generally positive human trial effects. Vitamin A’s efficacy varied, with positive monotherapy or adjunct effects. Vitamins B and D showed therapeutic benefits. Oral cancer research was extensive, with a focus on oral lichen planus and oral leukoplakia among the 11 OPMDs. All bias levels were reported in ‘selective reporting’ and ‘performance’, except for “definitely high” in the ‘selection’, ‘detection’, and ‘attrition/exclusion’ domains. Conclusions: Evidence of vitamin interventions for OPMDs and OC ranges from mixed to promising. Standardizing the study design and outcomes would enhance future research.

## 1. Introduction

Micronutrients, including vitamins, trace elements, and cofactors, are widely acknowledged as vital for cell and tissue homeostasis, integral to human nutrition [1]. Imbalances, either in deficiency or excess, can lead to oral mucosal and dental hard tissue pathologies [1,2].

These intricate constituents play pivotal roles in enzymatic reactions, cellular metabolism, and oxidative defence mechanisms [1,3]. Nevertheless, an imbalance in trace elements can result in toxicity, emphasizing the importance of maintaining precise levels in the body [2]. Recognizing their diverse functions is essential for preventing pathologies arising from deficiencies or excessive intake, highlighting the necessity for a balanced dietary approach.

These vitamins and their synonyms or related terms that are included in the search string are listed in Appendix A.

Trace elements are chemical micronutrients that play an important role in maintaining the integrity of physiological and metabolic processes within living tissues [2]. Nonetheless, trace elements can also exert toxicity if they are present in excess in the body, highlighting the importance of a homeostatic status of the trace elements [2]. After the initial search, the study selection criteria were narrowed to exclude trace elements due to the large number of studies identified. Thus, this report will focus on vitamins only.

Oral cancer (OC) includes malignant neoplasms originating from oral tissue sites including the inner lip, tongue, gingiva, floor of mouth, and palate [4]. Oral cancer constitutes 3% of all clinically diagnosed cancers with an age-standardised rate of 12.8 per 100,000 person-years nationally in Australia [5]. Despite several malignant neoplasms with greater prevalence, oral cancer screening by oral examination remains a vital element in patient prognosis due to the disease’s high mortality and morbidity [5]. The 5-year survival of oral cancer is approximately 50% and disability-adjusted life years (DALYs) have increased by 87.1% from 1990 to 2017 [6]. These are exacerbated by the risk factors including tobacco and alcohol consumption, and HPV infection. 

Oral potentially malignant disorders (OPMDs) are dysplastic conditions of the lip or oral cavity with implicated increased risk of malignant transformation [7]. Based on the consensus from the 2020 workshop organised by the World Health Organisation (WHO) Collaborating Centre for Oral Cancer, the list of OPMDs and their definitions are included in Appendix A. Notably, OPMDs share several risk factors with OC including smoking and alcohol; given the increased risk, timely screening, early identification, and diagnosis by oral health practitioners of suspicious lesions are important for timely care and patient prognosis, particularly in high-risk populations [7].

Previous research has identified several potential mechanisms for the anticancer properties of vitamins at a cellular level in preclinical studies, ranging from upregulating inhibitory growth signalling pathways and antioxidant properties to angiogenesis inhibition [8]. However, the translational benefits of using vitamins as interventions in clinical cancer studies show mixed evidence regarding their therapeutic benefit, efficacy, reproducibility, and tendency to produce adverse outcomes [8]. Furthermore, previous reviews have highlighted the impact of vitamins on oral health, highlighting mixed therapeutic benefits on dental diseases such as gingivitis and periodontitis [9].

Despite numerous past primary studies investigating the effects of specific vitamins on OPMDs and OCs, no study to date has systematically reviewed the effects and implications of vitamins in OPMDs and OC. In our study, we have explored the current evidence base to investigate which vitamins could potentially be beneficial for each specific OPMD and OC. We also sought to understand if essential vitamin supplementation offers therapeutic benefits for OPMDs and OC. The review has shown that vitamins C and K are understudied, while studies focused on oral lichen planus (OLP) and oral leukoplakia (OL), in addition to oral cancer. Overall, there was generally mixed to promising evidence for the therapeutic effects of vitamins on OPMDs and oral cancer.

## 2. Materials and Methods

The screening and data collection in this systematic review adhered to the 2020 version of the Preferred Reporting Items for Systematic Reviews and Meta-analyses (PRISMA) guidelines [10]. A comprehensive search was conducted on 16 May 2023 on the following databases:MEDLINEEvidence-Based Medicine (EBM)Web of Science (WOS)

The inclusion criteria specified studies that assessed the effect of essential vitamins and trace elements as interventions on:Human subjects with any relevant OPMD or OC, orIn vitro and in vivo OPMD or OC models

Relevant OPMDs were designated by Warnakulasuriya et al. [7] as shown in Appendix A. No restrictions were placed on publication date, geographical location, patient age, or gender. All articles released after 16 May 2023, were excluded. Meta-analyses, systematic reviews, other reviews, non-peer-reviewed articles, expert opinions, published extracts and letters to editors, non-English language articles, retracted articles, and studies using only self-reported measures were manually excluded. 

The initial aim of the project was to assess, if any, the therapeutic role of essential vitamins and trace elements in OPMDs and OC. The search strategy revolved around 3 core concepts:VitaminsTrace elementsOPMDs and OC

As such, the search string was implemented as per Appendix A. These records were imported into Covidence (Veritas Health Innovation, 2023, Australia) where duplicates were automatically removed.

Following the PRISMA protocol, two separate reviewers were chosen to conduct title and abstract screening independently in Covidence, according to the inclusion and exclusion criteria above. Pilot screenings of 30 and 50 articles were conducted, with conflicts discussed to ensure inter-examiner consistency prior to screening all records. The full-text screening conflicts were resolved by consensus or by a third group member if no consensus was achieved. Cohen’s kappa coefficient of inter-rater reliability was calculated at each step of screening (Appendix A). 

The number of articles remaining for full-text review was deemed unachievable for the time frame of this assessment; hence a new round of title/abstract screening was conducted to remove trace elements. These articles were set aside for a future comprehensive systematic review.

The vitamin-only articles were sent for full-text retrieval, with those unretrievable being excluded. Full-text review was performed based on independent teams of reviewers.

Each group member extracted records into a pre-determined data extraction table in Excel (Microsoft, 2023, Redmond, WA, USA). The domains captured included the following: study type, vitamin of intervention, presence of additional therapy, route of administration, dose, frequency of administration, and duration. Information specific to in vitro studies included cell origin, type, and organ. In vivo animal studies included animal strain, origin, gender, age and weight, and route of administration. Finally, in vivo human studies included country, sex, sample size, age, comorbidities, risk factors, and route of administration.

A quality assessment of the included articles was carried out using criteria adapted from the OHAT Risk of Bias Rating Tool for Human and Animal Studies (National Toxicology Program, 2015). For this study, five domains (selection, selective reporting, performance, attrition/exclusion, and detection) were assessed using eight questions, with modifications made to ensure relevance to in vitro studies. For each question, bias was assigned as: “definitely low risk”, “probably low risk”, “probably high risk”, or “definitely high risk”. The assessment questions are supplied in Appendix A.

The evidence compiled from the final studies included was presented in data extraction tables (Appendix A). Each domain was then sorted to analyse trends.

## 3. Results

### 3.1. Data Collection

A total of 4304 records were retrieved collectively from Medline, Embase, and Web of Science, with 634 duplicates automatically removed. After screening 3670 articles by title and abstract, 485 studies were included; Cohen’s kappa coefficients are described in Appendix A. From the remaining 209 vitamin-only full-text records, 80 records were analysed (Figure 1).

### 3.2. Quality Assessment of Studies (OHAT)

Two independent reviewers assessed all included studies using OHAT for risk of bias. An initial risk of assessment was performed upon a small selection of papers and conflicting bias allocations were resolved by discussion before subsequent assessment of all remaining papers. A “definitely low risk of bias” was observed in the categories of ‘selection’, ‘selective reporting’, ‘performance’, ‘attrition/exclusion’, and ‘detection’ (8.125%, 77.500%, 16.875%, 18.750%, and 4.375%, respectively). “Probably low risk” was observed in ‘selection’, ‘selective reporting’, ‘performance’, ‘attrition/exclusion’, and ‘detection’ (11.875%, 16.250%, 6.875%, 6.250%, and 8.750%, respectively). “Probably high risk” of bias was found in ‘selection’, ‘selective reporting’, ‘performance’, ‘attrition/exclusion’, and ‘detection’ (61.250%, 5.000%, 45.625%, 56.250%, and 86.875%, respectively). A bias rating of “definitely high risk” was observed in the ‘selective reporting’ and ‘performance’ domains (1.250% in both domains). The complete assessment of studies can be found in the Appendix A. 

### 3.3. Characteristics of Included Studies

The 80 studies included were published from 1959 to 2022, with the majority being produced between 1980 and 1990. Studies were contributed by 19 countries, with the USA having the most publications (Appendix A). The distribution of OPMDs and OC investigated according to each vitamin class is shown in Figure 2. The B complex vitamins have been collated as “Vitamin B” studies. A total of 21 studies have included vitamins as an adjunct. The distribution of study types is shown in Appendix A. Hybrid studies were defined as those incorporating multiple study designs.

#### 3.3.1. In Vivo Human Studies

A total of 35 in vivo human studies were identified and are summarised in Appendix A. The studies were conducted from 1962 to 2022 with the majority being published within the last 30 years. The included in vivo human studies explored OL, OLP, SF, DYS, and OC. 

Vitamin A was investigated in 25 studies either as a monotherapy or therapeutic adjunct. Vitamin A was delivered topically, orally, or systemically via intramuscular injections. Primarily studied in patients with OL (Figure 2), vitamin A reduced the number and size of OL lesions and decreased the level of perceived pain [11,12,13,14,15,16,17] (Four studies reporting vitamin A’s effects on lesion regression showed complete remission [18,19,20,21], while two studies did not observe a significant clinical remission [22,23]. Mixed evidence from vitamin A-treated OLP lesions was shown in two studies reporting reduced lesion number or improved clinical symptoms [24,25]; one study reported no change in lesion count [26]. Vitamin A was found to increase retinoic acid-binding protein expression in OC cells [27]; however, two other publications reported no statistically significant effect on the recurrence and death rate of OC [28,29]. Studies investigating vitamin A supplementation on SF and DYS reported improved burning mouth symptoms and improved mouth opening [30].

One study showed that vitamin B12 as a monotherapy and in conjunction with levamisole had reduced pain in patients and exerted a therapeutic effect on the ulcerative lesions in OLP [31]. Two studies including vitamin B with other vitamins also showed therapeutic effects on clinical symptoms of OLP and SF [1,26].

Vitamin C was used in three studies involving OLP, SF, and OL. The studies used vitamin C as part of their multi-therapy regimen. Oral administration of vitamins A and C found no significant results in the remission or malignant transformation of OL lesions to OC [23]. 

Three studies found that oral or topical vitamin D exerted a therapeutic effect on OLP through reduction in lesion number, decreased lesion severity, or improvements to clinical burning symptoms [32]. The role of vitamin D in SF was part of a multi-vitamin regimen and saw clinical improvement in SF symptoms [1].

Vitamin E was explored in seven studies looking at OLP, OL, OC, and SF. One study found that the topical application of vitamin E reduced the lesion size, without any therapeutic effects on the clinical symptoms of OLP. Two of three OL studies found that vitamin E applied orally or topically reduced lesion sizes, and histological changes were also observed [33,34]. One SF study utilized topical vitamin E with a corticosteroid (Betonil) and found that vitamin E enhanced the therapeutic effect of the corticosteroid [35]. In OC, the use of vitamin E with radiotherapy saw a reduction in markers correlating with malignant transformations [36]. There were no in vivo human studies concerning vitamin K.

#### 3.3.2. In Vivo Animal Studies

A total of 29 in vivo animal studies were identified, conducted between 1960 and 2020 (Appendix A). Among these 29 publications, 22 studies utilised the golden hamster animal model. A total of 26 studies focused on OC; the rest focused on oral lichen planus and oral leukoplakia. 

Ten studies on OC showed that vitamin A supplementation, oral or injected, significantly reduced the lesion size progression and tumour count, with signs of total inhibition of carcinogenesis in the hamster cheek pouch model [37]. Notably, one study reported that topical vitamin A treatment resulted in oral leukoplakia progressing to OSCC in the hamster buccal pouch model [38].

Furthermore, four studies described that vitamin C applied orally or topically restricted carcinoma growth and invasion into the sub-epithelium, and decreased the incidence of OL lesions [39,40,41]. However, one study reported that vitamin C supplementation resulted in significantly larger tumours and a slightly increased number of gross tumours [42]. 

Three of the six studies investigating vitamin D employed the administration of vitamin D via intraperitoneal injection, while the other studies administered vitamin D orally or topically. These studies reported that vitamin D supplementation significantly reduced the degree of dysplasia, and suppressed OC tumour growth in both mice and hamster study models [43].

Seven studies concluded that vitamin E applied orally or topically in the hamster buccal pouch model inhibited and prevented carcinogenic action in OC. The sole vitamin B study found that a low-vitamin B1 diet in hamster models resulted in malignant neoplasms arising in a significantly shorter period compared to hamsters with adequate vitamin B1 [44]. There were no in vivo animal studies concerning vitamin K.

#### 3.3.3. In Vitro Studies

A total of 23 in vitro studies were identified (Appendix A). They encompassed a variety of samples, including primary specimens from commercial cell lines and human and animal biopsies. Both cancer cells and dysplastic cells were commonly used. Five studies showed that vitamin A did not influence cell proliferation but may affect oral cancer progression through increased retinoic acid-binding protein expression, clonogenic activity, as well as potential protection against genotoxic damage. Concerning vitamin E, low-dose physiological concentrations were shown to support OSCC growth, while high-dose pharmacological concentrations inhibited cell growth, even when used as a single therapy [45]. Vitamin E was also shown to enhance the cancer-inhibitory effect of 13-cis-retinoic acid, interferon-alpha2A, and cisplatin [46].

One study discovered that vitamin B increased OSCC proliferation in a dose-dependent manner, and another concluded that vitamin B9 antimetabolites can be used as an alternative therapy for OCs that are resistant to other therapies.

Ten studies explored the effect of vitamin D on OC; generally, vitamin D administration was found to inhibit SCC growth and OLP development in a dose-dependent manner. When used as an adjunct along with soy isoflavone or cisplatin, vitamin D was shown to enhance the anti-tumour effect of other additional therapies.

The cytotoxic and antioxidant properties of vitamin K3 were highlighted in two papers, which were attributed to its anti-neoplastic and anti-migratory effects on precancerous cells, preventing progression to oral cancer. There were no in vitro studies concerning vitamin C.

## 4. Discussion

This review identified many studies that attempted to elucidate the role of vitamins in preventing or mitigating the effects of OPMDs and OC. Our review showed that the majority of the evidence involved specific vitamins (A, D, and E) and their role in select OPMDs (OL and OLP) and OC. In animal models, specifically hamster cheek pouches, the role of vitamins in preventing OC progression was promising [47,48,49,50]. However, the evidence of vitamins in OC upon translation into human clinical trials remains mixed: some studies found that Vitamin A did not significantly achieve a therapeutic effect [28,29], while supplementation of Vitamin E was shown to reduce markers of malignant transformation [36]. Several in vivo animal studies demonstrated the therapeutic effects of vitamin A on the progression of OCs; however, few reported outcomes regarding OPMDs.

Vitamin A preclinical studies predominantly showed therapeutic properties through its inhibitory effects on tumour cell growth and differentiation [26,27]; however, one in vitro study demonstrated no effect on preventing the progression of OLP into OC [48]. In clinical studies investigating OPMDs, reduction in lesion size and number, and improvement in patient symptoms in vitamin A adjunct or monotherapies were observed [21,51,52].

In clinical investigations, vitamins of the B complex used as part of a multi-therapy approach have been documented to exhibit therapeutic effects on the clinical symptoms of OLP and SF [1,26]. Additionally, preclinical studies have shed light on the potential repercussions of vitamin B deficiency in hamsters, highlighting an increased susceptibility to malignant neoplasms [44]. Notably, an in vitro study reported that vitamin B9 antimetabolites held promise as an alternative therapeutic option for OC that demonstrated resistance to conventional treatment modalities [53].

Vitamin C studies in three human trials aimed at conditions like OLP, SF, and OL, utilising vitamin C in combination therapies [1,23,26]. Vitamin C administration in animal studies showed positive effects on OPMDs and OC, inhibiting carcinoma growth and reducing oral leucoplakia lesions, except for one study that reported conflicting results of increased tumour size and number with vitamin C supplementation [39,40]. Unfortunately, no in vitro studies were included to report the effect of vitamin C on OPMDs and OC.

Several in vitro studies on vitamin D showed consistent findings supporting its potential to inhibit SCC growth and OLP development, and enhance the therapeutic effect of additional cancer therapies [54,55]. In vivo animal studies further supported its effects in delaying cancer progression [43], but showed no significant effect on oral squamous cell carcinoma cell death rate. However, there was increased cell death in a combination treatment with cisplatin and erlotinib. Therefore, vitamin D may have a potential inhibitory effect on cancer cell growth but can be suggested as an adjunct in cancer therapy rather than as a single treatment. When translated to human studies, vitamin D was investigated as an intervention only for OLP. With topical or oral supplementation of ~50,000 IU, its therapeutic effects worked to reduce the severity, lesion size, and burning sensation of OLP.

Vitamin E in vivo animal studies showed some extent of delaying and inhibiting effect on OC, with the results obtained from the buccal mucosa of hamsters [56]. One human study conducted topical vitamin E application on patients with OLP, showing decreased lesion size, which had also been identified in a study with OC patients [51]. Interestingly, with regard to the dosage of vitamin E, there seemed to be a dual effect with treatment, promoting OC growth at a low dosage (10 µM), while having a regulatory effect at a high dosage (100 and 154 µM) [54].

The current research landscape in the field of vitamins predominantly revolves around the exploration of the therapeutic effects and mechanisms of action associated with vitamins A, D, and E. There remains a lack of investigations on the potential therapeutic applications of vitamin K. Additionally, there is limited research examining the role of vitamins in the context of other OPMDs including AK, cGVHD, DC, EL, OLE, OLL, PVL, and PLRS. As some promising evidence is found, this may warrant further investigation of vitamins to broader types of OPMDs to enhance our understanding of their potential implications and therapeutic value to patients.

However, this review has several limitations: firstly, vitamins have diverse alternative nomenclature in the literature that our search strings have strived to incorporate. Nonetheless, due to the diversity of both synonyms and derivatives of each vitamin, active metabolites and synthetic forms, we acknowledge the possibility of relevant articles not captured in our study. Additionally, the heterogeneity in study characteristics, methods, and outcome measurements, makes direct comparisons of study findings onerous and may potentially impact the overall conclusions. Future studies may benefit from utilising a standardised guideline to report robust and comparable data surrounding this topic.

## 5. Conclusions

In summary, this comprehensive review elucidates the intricate roles of vitamins, particularly A, D, and E, in the context of oral potentially malignant disorders and oral cancer. The assessment of the therapeutic benefits of different vitamins can potentially guide clinicians in treating patients with OPMDs and oral cancer. While preclinical studies in animal models demonstrate promising therapeutic properties, clinical translation reveals mixed outcomes, highlighting the need for further research and nuanced approaches. Vitamin A exhibits inhibitory effects on tumour cell growth, yet its efficacy in preventing oral cancer from progressing remains debated. The vitamin B complex, when integrated into multi-therapy approaches, shows therapeutic promise in alleviating clinical symptoms of OPMDs. Vitamin C, employed in combination therapies, exhibits positive effects on OPMDs and OC in animal studies; however, conflicting results from one study warrant further investigation. Vitamin D consistently demonstrates potential in inhibiting squamous cell carcinoma growth and enhancing cancer therapy effects. Vitamin E, both in animal models and human studies, presents dual effects contingent on dosage, emphasizing the need for precise administration. Notably, the research landscape primarily centres around vitamins A, D, and E, leaving vitamin K and other OPMDs warranting further exploration. Despite limitations in nomenclature and study heterogeneity, this review provides crucial insights, emphasizing the imperative of standardized guidelines for future research in this vital area of study.

## Figures and Tables

**Figure 1 jpm-13-01520-f001:**
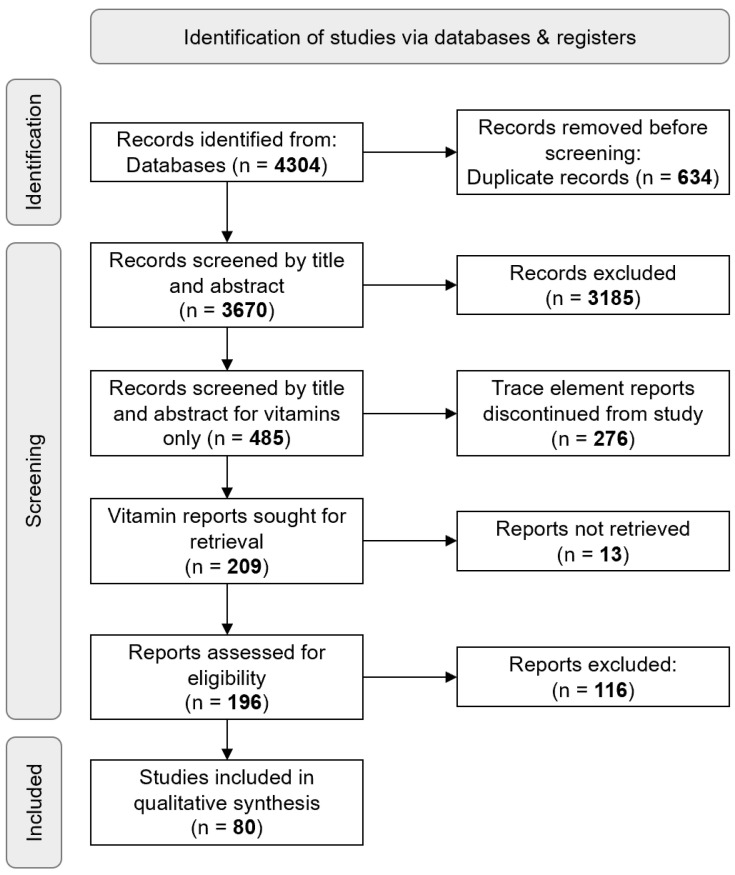
PRISMA flow chart showing the process of literature search and screening.

**Figure 2 jpm-13-01520-f002:**
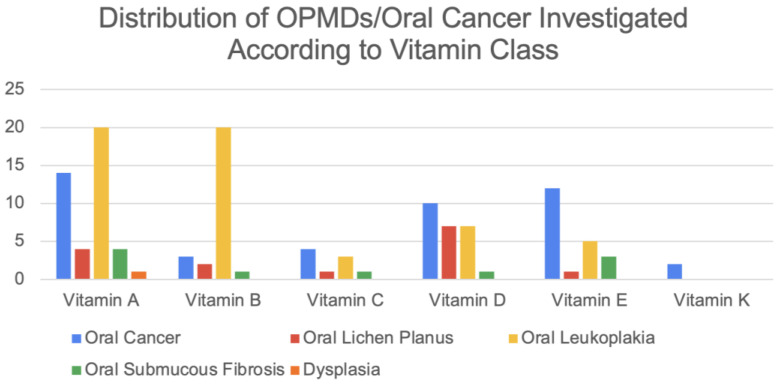
The distribution of OPMDs/oral cancer investigated according to each class of vitamin.

## Data Availability

The data that support the findings of this study are available from the corresponding author upon reasonable request.

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
