# Peer review of "The Role of Vitamins in Oral Potentially Malignant Disorders and Oral Cancer: A Systematic Review"

_jpm, 2023, doi:10.3390/jpm13101520_

Round 1
Reviewer 1 Report
Dear Authors,
The manuscript is well written, presents itself in a structured manner, and attempts to summarize the wealth of previous findings on the therapeutic use of various vitamins.
This is an interesting and potentially valuable study but the manuscript could be improved in some ways. A number of minor critical points are described below:
Title
The title should be reconsidered and trace elements removed, as despite being mentioned at the beginning of the review, they were omitted from further evaluation and are no longer considered.
Results
[P4, L150] It is mentioned that 81 studies were analyzed, but in the further course only 80 are mentioned. Is this a typing error?
Author Response
Reviewer 1
Dear Authors,
The manuscript is well written, presents itself in a structured manner, and attempts to summarize the wealth of previous findings on the therapeutic use of various vitamins.
This is an interesting and potentially valuable study but the manuscript could be improved in some ways. A number of minor critical points are described below
Reply: Thank you for this comment. We are very glad this reviewer found the manuscript valuable, easy to read, and with high translational value.
Comment: Title: The title should be reconsidered and trace elements removed, as despite being mentioned at the beginning of the review, they were omitted from further evaluation and are no longer considered.
Reply: Thank you for this comment. We corrected the title as suggested and now it reads: “The Role of Vitamins in Oral Potentially Malignant Disorders and Oral Cancer: A Systematic Review”
Comment: Results, [P4, L150] It is mentioned that 81 studies were analyzed, but in the further course only 80 are mentioned. Is this a typing error?
Comments and Suggestions for Authors
Reply: Thank you for this comment. Yes, the studies included and analyzed were 80, we have then corrected this typo as suggested.
Reviewer 2 Report
Dear editor and authors,
This was a well-prepared review which evaluated and summarised the potential role of different essential vitamins on various OPMDs and OC. From the conclusions, Also, the authors recommended different vitamins can potentially guide clinicians in treating patients with OPMDs and OC. Future research methodology can benefit from systematically reporting the potential therapeutic effect of each vitamin while encompassing a broader range of vitamins and OPMDs.
Dear editor and authors,
This was a well-prepared review which evaluated and summarised the potential role of different essential vitamins on various OPMDs and OC. From the conclusions, Also, the authors recommended different vitamins can potentially guide clinicians in treating patients with OPMDs and OC. Future research methodology can benefit from systematically reporting the potential therapeutic effect of each vitamin while encompassing a broader range of vitamins and OPMDs.
Author Response
Reviewer 2
Comments and Suggestions for Authors
Dear editor and authors,
This was a well-prepared review which evaluated and summarised the potential role of different essential vitamins on various OPMDs and OC. From the conclusions, Also, the authors recommended different vitamins can potentially guide clinicians in treating patients with OPMDs and OC. Future research methodology can benefit from systematically reporting the potential therapeutic effect of each vitamin while encompassing a broader range of vitamins and OPMDs.
Comments on the Quality of English Language
Dear editor and authors,
This was a well-prepared review which evaluated and summarised the potential role of different essential vitamins on various OPMDs and OC. From the conclusions, Also, the authors recommended different vitamins can potentially guide clinicians in treating patients with OPMDs and OC. Future research methodology can benefit from systematically reporting the potential therapeutic effect of each vitamin while encompassing a broader range of vitamins and OPMDs.
Reply: Thank you for this comment. We are very glad this reviewer found our manuscript valuable, and no further corrections needed were requested.
Reviewer 3 Report
I applaud the authors' effort in this systemic review of the role of vitamins and trace element therapies in OPMD and OC. The conclusion is robust based on their extensive review of the existing literature.
Author Response
Reviewer 3
Comment: I applaud the authors' effort in this systemic review of the role of vitamins and trace element therapies in OPMD and OC. The conclusion is robust based on their extensive review of the existing literature.
Reply: We sincerely appreciate the reviewer’s kind words and recognition of our diligent work in conducting this comprehensive systematic review on the role of vitamins and trace element therapies in OPMD and OC; your positive feedback further validates the strength of our conclusions derived from the extensive examination of the existing literature.